# Analysis of stability law and optimization of slope angle during excavation of deep concave mine slope

Lili Wu[1]*, Keqiang He[1], Lu Guo[2], Linna Sun[1]

**1** Department of Civil Engineering, Qingdao University of Technology, Shandong, China, **2** Department of Civil Engineering and Architecture, Suqian University, Jiangsu, China

* wll1603863477@163.com

## Abstract

Occasional collapse failure is a typical occurrence during mine slope excavation processes. This study aimed to investigate the disaster law in the process of mining slope excavation, and further explore the optimal selection of excavation angle. Based on the systematic analysis of the residual sliding force and deformation response characteristics during slope excavation, the increment of the residual sliding force increases and the stability coefficient decreases with the increase in excavation depth. Additionally, a numerical model of the Jinchuan mining area in Jinchuan City, Gansu Province, China was created using the Midas-GTS finite element software. The influence of different excavation slope angles on slope stability was analyzed via numerical simulation under certain step slope height and width. The results show that the force and deformation of the slope were unfavorable to slope stability, and the slope stability coefficient would decrease gradually with the increase in slope angle. In addition, the optimal excavation angle combination ranges were determined as 62˚~ 65˚, 64˚~ 67˚, 67˚~ 69˚, 70˚~ 71˚, 73˚, 75˚~ 76˚, 77˚~ 80˚considering the stability and maximum recovery. Therefore, the above research results verify the loading effect of mine slope excavation, and can serve as a reference for studies on the optimal range of excavation angles for the mine slope.

## Introduction

In mining engineering, open pit excavation is a common mining method. The stability variation law and excavation angle in the mining process are two important contents in the field of deep concave mine slope. Owing to the complex geological conditions, the mining area in Shaanxi Province, China, is at considerable risk of collapse. Sliding may occur during the excavation, and the maximum vertical displacement is 2.1 cm [1]. In the Xincheng gold mine project in China, the rock mass has an obvious displacement due to the excavation of the mine, which has a serious impact on the safety of the mining process [2]. In addition, water outbursts and rock bursts often occur during the mining of coal seams under the water-rich sandstone strata with thicknesses that exceed 50 m, otherwise called ultrathick-and-hard strata (UTHS),

**Data Availability Statement:** All relevant data are within the paper and its Supporting Information files.

**Funding:** This research was supported by the National Natural Science Foundation of China

(41372297) (KQ H), the Natural Science Foundation of Shandong Province (ZR2020KE004) (KQ H), the Open Fund of Key Laboratory of Geological Safety of Coastal Urban Underground Space, Ministry of Natural Resources (BHKF2021Y05) (L G).

**Competing interests:** No potential conflict of interest was reported by the authors.

which are common in mining regions of northwestern China [3]. The Higashi-Shikagoe limestone quarry is an open-pit mine situated in the Hokkaido Prefecture, Japan, that has experienced four slope failure incidents since 1996 [4]. In excavation engineering, by analyzing the stability of a group of headings driven in the high horizontal stress fields in the copper ore mines of the Legnica-Glogow Belt (LGCB), the driving direction were determined to be of key importance for the stability of the headings in LGOM mines [5–7]. As a high-efficiency and low-consumption mining mode, multi-middle section combined backfilling mining (MMSCBM) is becoming more and more widely used in metal mines. MMSCBM can effectively buffer the disturbance of the excavation to the surrounding rock, adjust the stress release mode, change the stress concentration area, and improve the stability [8]. From the perspective of geotechnical engineering, rock pillars can be defined as in-situ rock between two or more underground openings. However, in the analysis of hard rock pillars and in rock mass models to determine rock mass strength, it is very important to consider the actual stress level and stress path imposed on the pillars due to the excavation sequence of the pillars [9]. From the perspective of excavation stability, the complex nonlinear mechanical behavior of the combined system of rock and coal that surrounds mine openings is a major obstacle in the excavation, stability, and support design of mining thin coal seams [10–15]. The geological structure of the Changshanhao open-pit mine in Urad Middle Banner, Inner Mongolia, China is extremely complicated, and various slope instabilities have occurred, such as wedge sliding, bedding sliding, and toppling failure. Slope failure occurs to endogenic and exogenic integration, including physical and mechanical properties of the rock mass, geological structures such as faults and joints, and anthropogenic factors such as blasting and excavation disturbances [16]. In underground mining activities, rainfall and the dip of excavated rock slope also has a potentially important impact on the initiation and reactivation of slope deformation, especially on the steep rock slope [17–19].

In addition to the above aspects, numerical simulation analysis of the slope excavation process has considerable research significance. Meng et al. summarized the negative effect of excavation on the slope deformation and stability by numerical simulation [20]. Wang et al. studied the influence of mountain excavation on the redistribution of surrounding rock stress by means of numerical simulation [21]. Zhuo et al. used a numerical model coal seam and found that excavation increases the deformation and affects the safety of the coal mine [22]. Considering the excavation in the Xiamen Haicang tunnel excavation as a research object, Wang et al. used the FLAC3D numerical simulation software to study the temporal and spatial effects of the complex excavation process to understand the influence of construction activity on stability [23]. By means of the finite element numerical simulation method, Ding et al. analyzed the influence of excavation loading and water softening on rock mass and verified the effectiveness of the simulation method [24]. Through geological analysis and numerical simulation, Chen et al. quantitatively understood the impact of excavation on the karst groundwater and geological environment [25]. Liu et al. carried out the numerical simulation on the excavation of a certain foundation pit and found that the results of the numerical model are consistent with the actual deformation of the foundation pit [26]. Ze et al. used numerical simulation to determine the influence of multi-step excavation on the surrounding rock deformation, and the internal displacement was long lasting [27]. The above studies have routinely analyzed the stability law of mine slopes, but the dynamic mechanism of the excavation instability of such slopes have not been studied as extensively.

Considering the large quartz open-pit mine of the Jinchuan Group in Gansu Province in China as an example, the influence of different excavation angles on slope stability is analyzed in order to obtain the optimal combination of excavation angle parameters and provide references for the construction of related projects.

## Study on the influence law of the excavation loading process on slope stability in deep concave mine

### Evolution characteristics of slope sliding dynamics and deformation response in deep concave mines

During slope excavation in deep pit mines, the initial stress equilibrium state in the slope body is affected by the excavation behavior, and stress redistribution occurs [28–31]. Stress redistribution is a typical phenomenon associated with excavation-induced slope failures. Fang et al. summarized the stress redistribution characteristics and deformation failure mechanism of the slope under excavation condition [32]. Bu et al. established the redistribution equation of mining stress in coal mines and analyzed the continuous transmission of mining stress, which provides a reference for the safe and efficient mining of coal resources [33]. Fang et al. carried out physical model tests of arching slopes under centrifugal conditions and explored the stress redistribution characteristics of a weak claystone slope [34]. Yang et al. found that the in-situ stress redistribution in highly stressed rocks is a dynamic process that will result in a large damage zone [35]. In fact, the fundamental impact of excavation behavior is to increase the sliding force of the slope, which is equivalent to continuously loading the slope during the entire construction process. At the same time, there will be a corresponding displacement near the slope face as the excavation progress [36–38]. It is assumed that the slope of the deep concave mine is homogeneous, the homogeneous body is isotropic, and the thickness of the slope changes uniformly. Taking the potential sliding strip of slope as the research object, the stress of the sliding strip under the dynamic action of excavation is as follows (Fig 1).

Where $H_{i1}$ and $H_{i2}$ are respectively the height before and after the excavation of the sliding strip, m; $G_i$ and $G'_i$ are respectively the gravity before and after the excavation of the sliding strip, kN; $\Delta P_i$ and $\Delta P'_i$ are respectively the remaining sliding forces before and after the excavation of the sliding strip, kN; $R_i$ is the anti-sliding force of the sliding strip, kN; $N_i$ is the effective stress of the base, kN; $h'$ is the increased height of the sliding strip after excavation, m; $h$ is the excavation height of sliding strip, m; $\theta$ is the included angle between the sliding surface of the sliding strip and the horizontal plane.

Fig 1 shows that the sliding force and anti-sliding force are the gravity components of the slide block. The sliding force is represented by $T$, and the anti-sliding force is represented by $R$.

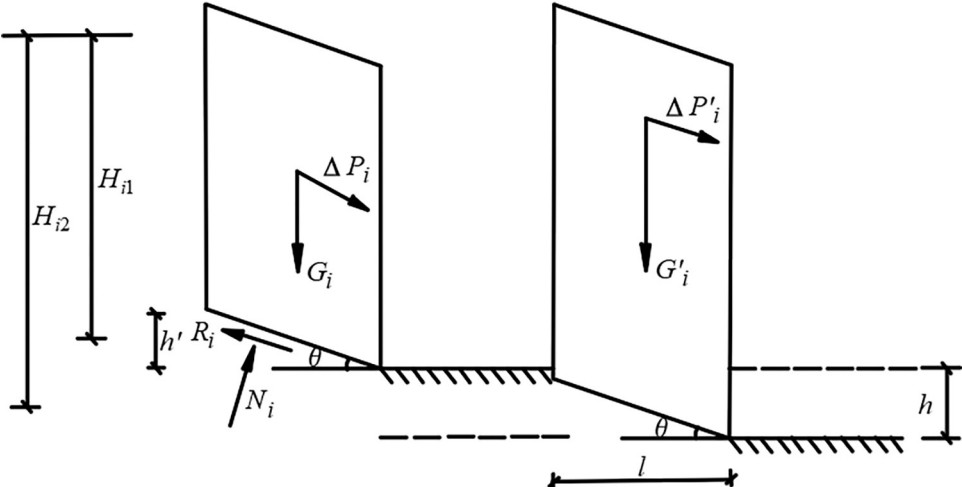

**Fig 1. Force analysis of sliding strip.**

The sliding force and anti-sliding force of the sliding strip before and after excavation are shown as follows:

Sliding force before excavation:

$$T_1 = G_1\sin\theta = (\gamma H_1)\sin\theta \tag{1}$$

Sliding force after excavation:

$$T_2 = G_2\sin\theta = (\gamma H_2)\sin\theta \tag{2}$$

Anti-sliding force before excavation:

$$R_1 = G_1\cos\theta\tan\varphi + cl = (\gamma H_1)\cos\theta\tan\varphi + cl \tag{3}$$

Anti-sliding force after excavation:

$$R_2 = G_2\cos\theta\tan\varphi + cl = (\gamma H_2)\cos\theta\tan\varphi + cl \tag{4}$$

Where $T_1$ is the sliding force before the excavation of the sliding strip, $kN$; $T_2$ is the sliding force after the excavation of sliding strip, $kN$; $R_1$ is the anti-sliding force of sliding strip before excavation, $kN$; $R_2$ is the anti-sliding force of the sliding strip after excavation, $kN$; $G_1$ is the gravity before excavation of sliding strip, $kN$; $G_2$ is the gravity after excavation of sliding strip, $kN$; $c$ is cohesion, $kPa$; $\varphi$ is the angle of internal friction,°; $\theta$ is the included angle of the sliding surface, °; $l$ is the bottom width of the slider block, $m$; $\gamma$ is the natural gravity of rock and soil mass, $kN/m^3$. Other symbols have the same meaning as above.

Therefore, the increment of sliding force of the sliding strip caused by excavation is:

$$\Delta P = P_2 - P_1 = (T_2 - T_1) - (R_2 - R_1) = (H_2 - H_1)(\gamma\sin\theta - \gamma\cos\theta\tan\varphi) \tag{5}$$

Where $P_1$ is the remaining sliding force before the excavation of sliding strip, $kN$; $P_2$ is the remaining sliding force after excavation of sliding strip, $kN$; $\Delta P$ is the sliding power increment of the sliding strip, $kN$; Other symbols have the same meaning as above.

Eq 5 shows that for specific slope, $(\gamma\sin\theta - \gamma\cos\theta\tan\varphi)$ is the constant value, and $(H_2 - H_1)$ is the increase in sliding strip height caused by excavation. Only when the slope is excavated downward, the volume of potential sliding soil will increase, and the height of the sliding strip will increase. Therefore, it can be considered that the excavation depth corresponds to the remaining sliding force increment in the process of slope excavation in mines with deep depressions.

For the calculations, the slope body is assumed to be ideal elastic-plastic, and the stress and deformation physical parameters of the slope body section are averaged [39]. According to the basic principle of elastic-plastic mechanics, the relationship between the sliding force and displacement variation caused by the excavation and loading of the slope in a deep concave mine is expressed as follows.:

$$\Delta S_i = l\bar{\varepsilon}_i \tag{6}$$

$$\bar{\varepsilon}_i = \frac{\bar{\sigma}_i}{\bar{E}_i} \tag{7}$$

$$\bar{\sigma}_i = \frac{\Delta P_i}{V_i} \tag{8}$$

Where $\Delta S_i \bar{\varepsilon}_i$ and $\bar{\sigma}_i$ are the displacement variation, mean value of strain and mean value of stress of sliding strip caused by excavation loading; $\bar{E}_i$ is the mean value of deformation modulus of sliding strip; $V_i$ is the volume of sliding strip (unit width condition), $V_i = H_i/\cos\theta$

Eqs 5 to 8 show that when the other factors remain relatively unchanged, the slope sliding dynamics and displacement variations caused by slope excavation in deep and concave mines mainly depend on the variations of excavation height. With the increase of excavation height, the sliding force and displacement change of the slope increase. Therefore, it can be considered that the excavation loading process is the main dynamic factor that affects the sliding dynamics and displacement variation of the mine slopes.

## Stability evolution of mine slope under excavation loading condition

According to the above equation and the limit equilibrium principle of slope stability, the stability coefficient of the mine slope after excavation is determined as follows:

$$F_{si} = \frac{(\gamma H_{i2})\cos\theta\tan\varphi + cl}{(\gamma H_{i2})\sin\theta} = \frac{\cos\theta\tan\varphi}{\sin\theta} + \frac{cl}{\gamma H_{i2}\sin\theta} \tag{9}$$

Eq 9 shows that the excavation height alone is the variation parameter in the process increase of mining slope loading, while other physical and geometric parameters are relatively constant. Therefore, the slope stability coefficient is directly related to the dynamic change in excavation loading.

## Study area

### Landslide situation

The Jinchuan Group open-pit quartz mine is located in Jinchuan District, Jinchang City, Gansu Province, China (Fig 2), and mining was started in 1980. The ore area is traversed by a series of SW-NE strike thrust faults (Fig 3), and the intrusive ore bodies are divided into several sections, which are divided into mining areas , , and by the Gansu Geological Survey Institute [40]. Among them, mine and are mined and managed by Longshou Mine of Jinchuan Company, mine is mined and managed by West No 2 Mine and mine is mined and managed by East No 3 Mine [41]. The mining distribution in the Jinshan mine is as follows (Fig 4). Up to now, the mining area has formed 1826 m, 1816 m, 1792 m, 1780 m, 1768 m, 1756 m, 1744 m, 1732 m, and other steps from top to bottom. With the development of mining, there have been many slope failures, such as local landslides on platforms 1816 and 1732 (Fig 5). Among them, the 1732 m platform exposed a fault zone, affecting slope safety. At the same time, due to prolonged mining, there have been local cracks on the surface of the deep concave mine slope (Table 1).

### Numerical simulation

According to the actual situation of the mining area, the section 5C~1C of the north slope of the mining area where local landslide occurred was selected as the modeling object, including seven steps between 1826 m and 1732 m (Fig 6).

According to the geological survey and test data provided by the mining area, the physical and mechanical parameters of the rock mass in the mining area are as follows (Table 2) [42].

According to the selected research object, four kinds of excavation slope angle parameters are set in the experimental scheme. The excavation angle increases gradually in schemes 1 to 2, and there is an overall steepening of the multi-step slope in schemes 3 to 4 (Table 3). Before

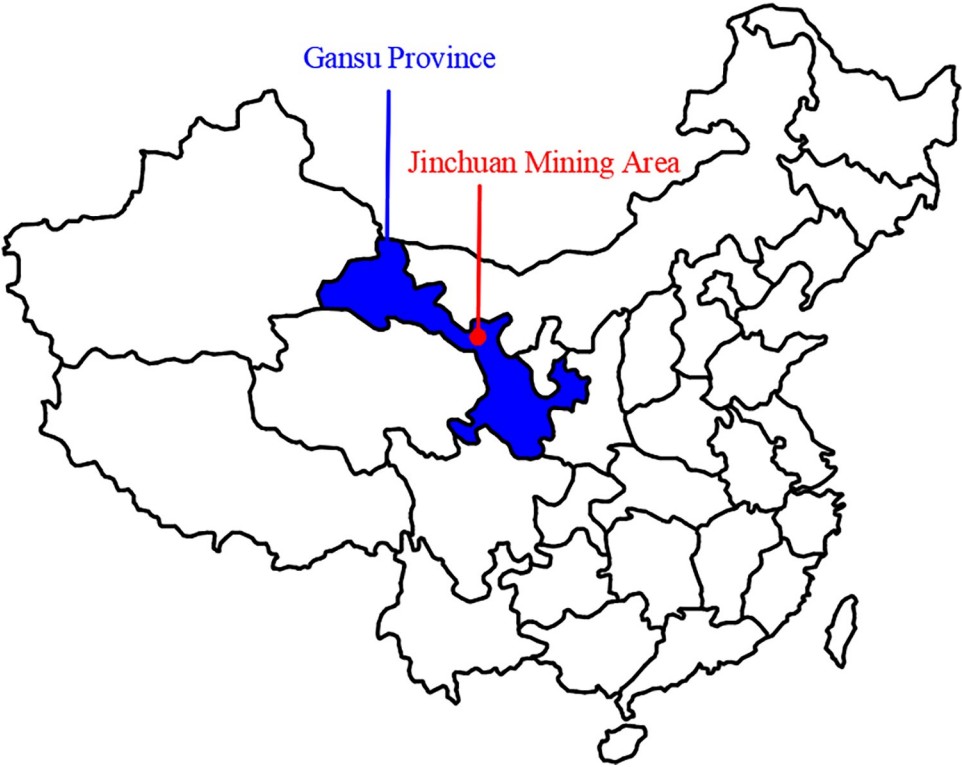

**Fig 2. Location of Jinchang City in China.**

the numerical simulation of the experimental scheme, the design scheme of the mine slope is simulated first, which is convenient for comparative analysis.

The Midas-GTS software was used to calculate the overall stability coefficient of slope for the design scheme and experimental scheme. Midas-GTS is a commonly used analysis

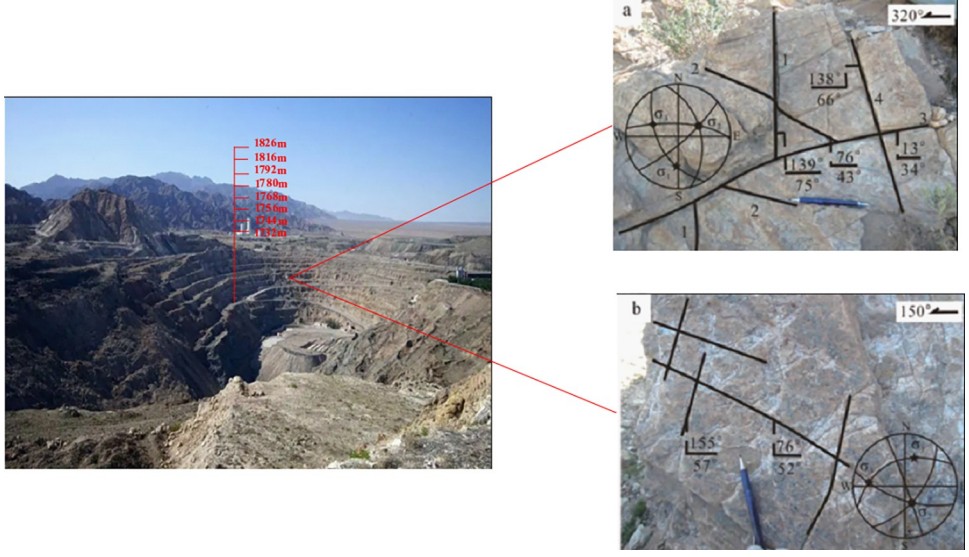

**Fig 3. Fault characteristics.**

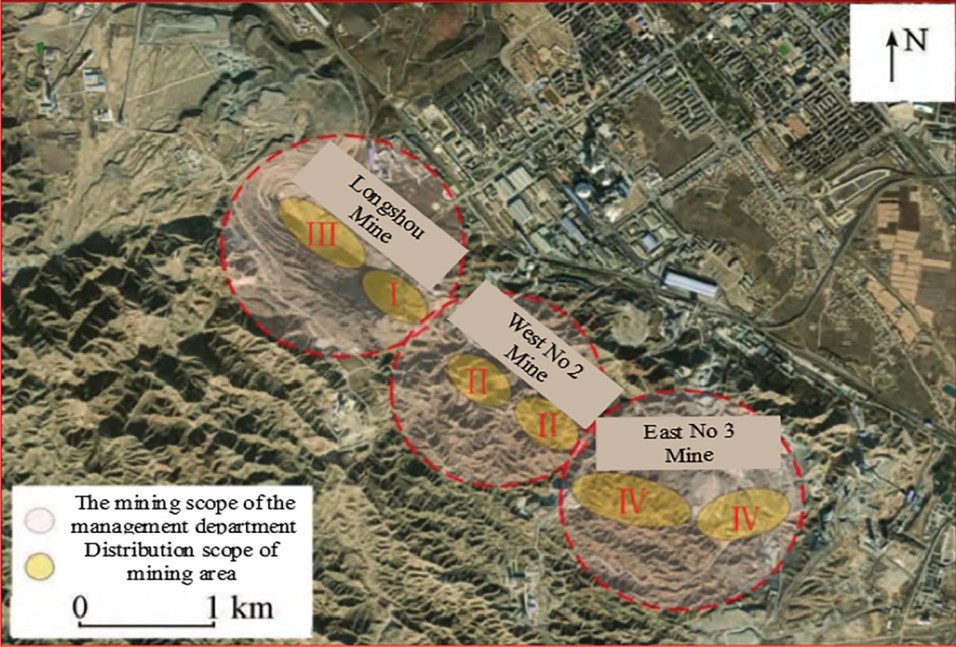

**Fig 4. Mining distribution map of Jinshan Mining area.**

software in the field of geotechnical engineering, that is capable of stress analysis, seepage analysis, slope stability analysis and other analysis types. Compared with other software, Midas-GTS software provides convenient geometric modeling and post-processing functions, which can be used extensively for underground structures, rock and soil, hydraulic, and other fields. Based on the calculations, the variation in stress, displacement, plastic zone distribution, and overall stability coefficient after excavation were compared and analyzed. By referring to the relevant specifications, the limit value of the slope stability coefficient is determined to be 1.3, and the stability coefficient of each scheme was compared with the specification value.

## Results analysis

### Analysis of design scheme

**Stress field analysis.** The stress cloud diagram during slope excavation is shown below. Before excavation, the horizontal stress in the slope is all compressive, and the stress field is

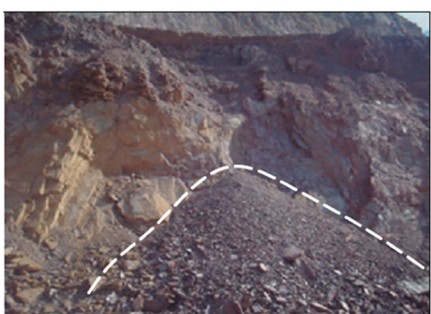
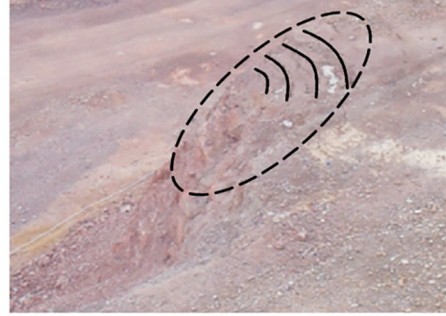

**Fig 5. Local failure of slope.**

**Table 1. Surface cracks and structural characteristics of rock mass.**

| Excavation steps (m) | Local crack condition (width, cm) | Geotechnical structure |
|---|---|---|
| 1826–1816 | 2–4 cm cracks | It is composed of Quaternary alluvium. |
| 1816–1792 | | |
| 1792–1780 | Tiny cracks | The integrity of rock mass is good. |
| 1780–1768 | | |
| 1768–1756 | 3–5 cm cracks | The rock mass is relatively broken, including faults and fracture zones. |
| 1756–1744 | | |
| 1744–1732 | | |

uniform strip distribution. When the slope is excavated to the 1732 m platform, the horizontal stress of the slope body is compressive, which increases with the slope depth. Stress concentration occurs at the foot of each step slope, and the stress in other parts is distributed in a uniform band (Fig 7). As in the distribution of horizontal stress field, the excavation also destroys the vertical initial stress state of the slope and causes stress redistribution. The vertical stress of the slope is also compressive, but there is no stress concentration. (Fig 8). Therefore, the vertical stress does little damage to the slope.

Considering that the horizontal stress has a great influence on the slope, the element located in the middle of each step and the top of the slope is selected in the model, and its horizontal stress value is extracted. Thus, the curve of the horizontal stress variation of each step during slope excavation is drawn (Fig 9). It can be seen from the Fig 9 that, as the excavation progress, the horizontal stress in the slope and the top has the same change trend, and the horizontal compressive stress of the lower step decreases gradually, with a decrease of approximately 50%. At the same time, the horizontal compressive stress of the first and second steps is also in a stable state without an increasing trend. In addition, for the same step, the value of the horizontal compressive stress at the top of the slope is less than that at the middle of the slope, because tensile failure usually appears at the top of the slope first, and because each step is in the "underground" state first, before being excavated to form a part of the slope. Therefore, the change in the early stage of the curve in Fig 9 shows an irregular rise and fall phenomenon, and the change trend is consistent after excavation.

**Displacement field analysis.** Fig 10 is the cloud map of the horizontal and vertical displacement during slope excavation. With the increase in excavation depth, the direction of horizontal displacement near the slope face changes gradually from the deviating position to one that faces the slope, and the value of horizontal displacement increases with the increase in slope depth. For each step excavated, obvious unloading rebound occurs on the step and at the bottom of the pit [43,44].

The element located in the middle of each step is selected in the model, and the displacement value is extracted to plot the curve of the displacement of each step in the process of slope excavation (Fig 11). As can be seen from the Fig 11, the step at the upper part of the slope has a negative displacement, and the displacement gradually increases and tends to be stable. The step at the lower part of the slope has a positive displacement, and the displacement value increases gradually. If the excavation continues downward, it can be predicted that when the negative displacement of the upper step of the slope tends to be stable, its value will gradually develop to the positive displacement, while the positive displacement of the lower step will continue to increase.

**Plastic zone and global stability analysis.** Depending on their distribution range, plastic zones are generated from the foot of each step slope, from which they start to develop upward,

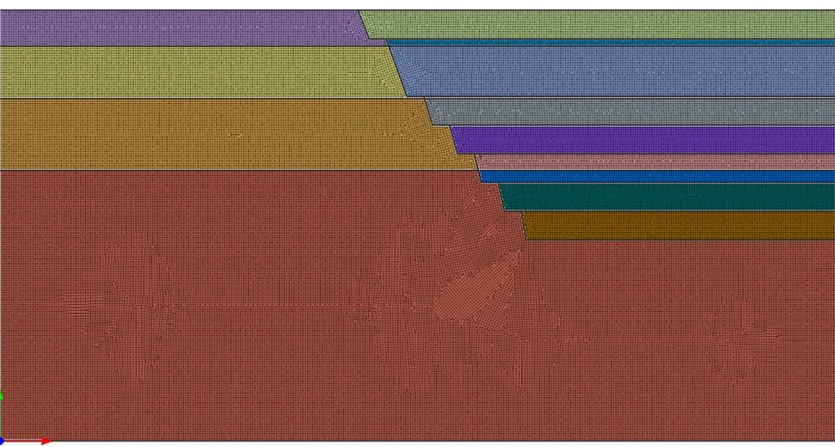
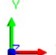

**Fig 6. Excavation model of slope.**

gradually connecting and penetrating (Fig 12). The stability coefficient of slope during excavation can be obtained by the strength reduction method (Fig 13). Fig 13 shows that the stability coefficient decreases as the slope is excavated. At the fourth step of excavation, the stability coefficient increases owing to the redistribution of the slope stress during the dynamic excavation. The cracks or shear failure arising from the previous excavation will close with the redistribution of stress, resulting in the establishment of a new stability of rock and soil weight. It can also be seen from the cloud map of the plastic zone that during the fourth excavation step, the range of plastic zone generated by the third and fourth steps is very small compared with the other parts. This also indicates that when the slope is excavated to the fourth layer, the stability coefficient is larger than that of the first three steps. However, in general, with the progress of excavation, the stability of the slope gradually decreases until the end of mining.

The element located in the middle and at the foot of each step is selected in the model, its maximum shear stress value is extracted, and the maximum shear stress curve of each step in the process of slope excavation is made. Fig 14 shows that with the progress of excavation, the variation trend of the maximum shear stress at the middle and foot of slope is the same. The shear stress of the first and second steps is relatively stable, while the shear stress of the other steps decreases before excavation. However, after excavation, the shear stress begins to increase and gradually tends to be stable. In addition, for the same step, the shear stress at the foot of the slope is greater than that at the middle of the slope, which is caused by the shear failure that usually appears at the foot of the slope first. It is worth noting that the shear stress curves in the early stage show a downward trend, as each step is "underground". However, after the excavation, the shear stress variation trend of each step is basically the same.

**Analysis of experimental scheme.** According to the simulation results of schemes 1~2, the distribution rules of the horizontal stress, horizontal displacement, vertical stress, and

**Table 2. Rock and soil parameters of slope [42].**

| Stratum | E (MPa) | c (kPa) | Φ (°) | γ (kN/m³) | μ | Thickness (m) | Density (g/cm³) |
|---|---|---|---|---|---|---|---|
| Quaternary | 1038.00 | 26.00 | 26.00 | 16.50 | 0.15 | 15.00 | 1.79 |
| Carboniferous-Permian System | 1520.00 | 109.40 | 29.75 | 26.20 | 0.14 | 22.00 | 2.19 |
| Upper formation of Lower Paleozoic | 1392.00 | 245.10 | 33.00 | 27.80 | 0.21 | 30.00 | 2.50 |
| Lower Paleozoic lower group | 1798.00 | 148.20 | 35.00 | 27.00 | 0.24 | / | 2.69 |

**Table 3.  The parameters of excavation slope angle in the experimental scheme (˚).**

| Schemes | Step 1826–1816 | Step 1816–1792 | Step 1792–1780 | Step 1780–1768 | Step 1768–1756 | Step 1756–1744 | Step 1744–1732 |
|---|---|---|---|---|---|---|---|
| **1** | 60 | 62 | 65 | 68 | 71 | 74 | 77 |
| **2** | 62 | 64 | 67 | 70 | 73 | 76 | 80 |
| **3** | 65 | 67 | 69 | 71 | 73 | 75 | 77 |
| **4** | 66 | 68 | 70 | 72 | 74 | 76 | 78 |

vertical displacement are similar to the design scheme. Among them, the horizontal displacement in schemes 1 and 2 changes with the slope angle, the negative horizontal displacement decreases, and the positive horizontal displacement increases gradually (Tables 4 and 5). This range gradually expands from the bottom of the slope to the top of the slope. The vertical displacement decreases gradually with the increase in slope angle, that is, the recovery value of the excavation unloading of rock mass decreases, but the reduction range is limited.

The distribution of plastic zone in schemes 1~2 is different from that in the design scheme: with the increase in slope angle, the plastic zone at the foot of the slope of steps 3~4 is gradually reduced compared with that in the design scheme, while the plastic zone at steps 5~7 is gradually increased compared with that in the design scheme. Therefore, there is a greater risk of local landslides at steps 5 to 7, which needs to be controlled during construction (Fig 15).

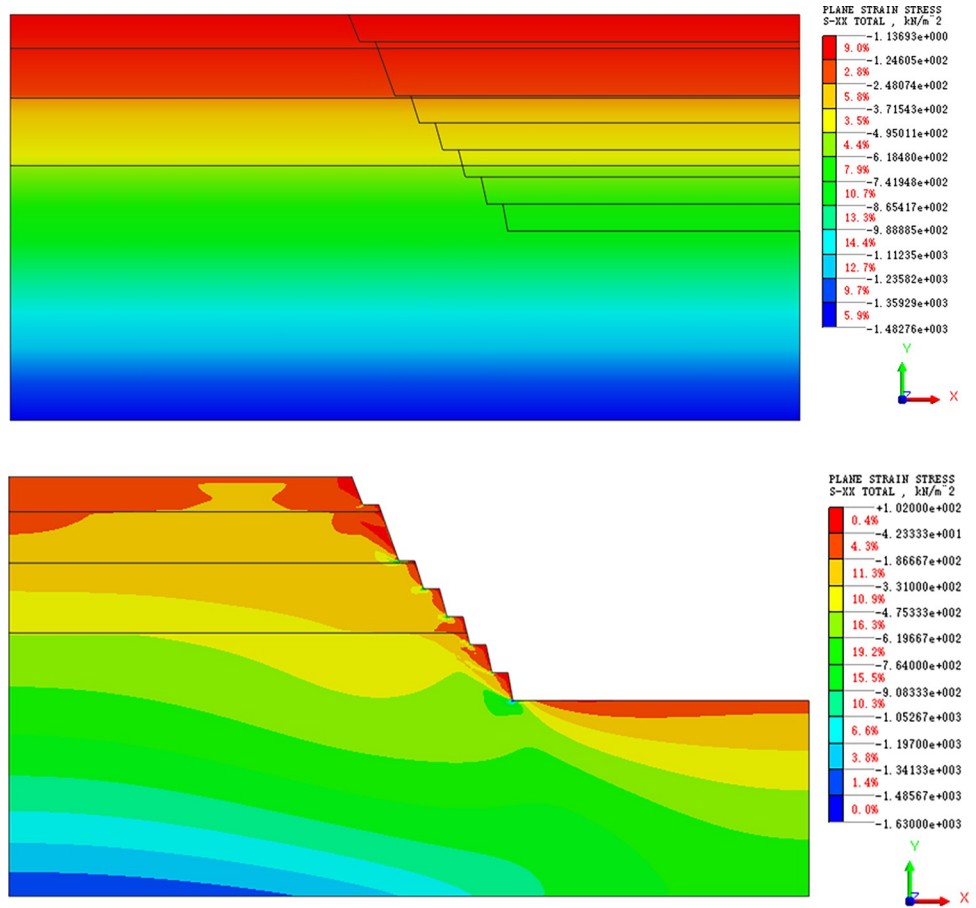

**Fig 7.**  (a). Horizontal stress cloud map (initial stress equilibrium). (b). Horizontal stress cloud map (excavation stage).

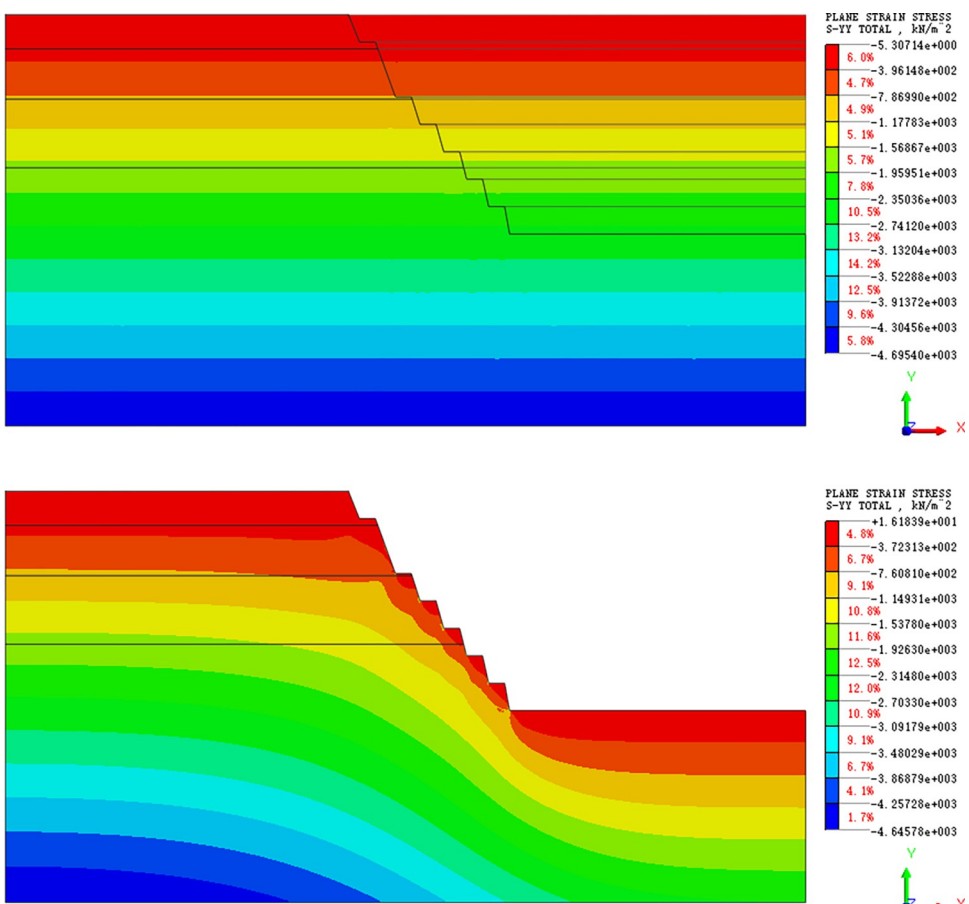

**Fig 8.** (a). Vertical stress cloud map (initial stress equilibrium). (b). Vertical stress cloud map (excavation stage).

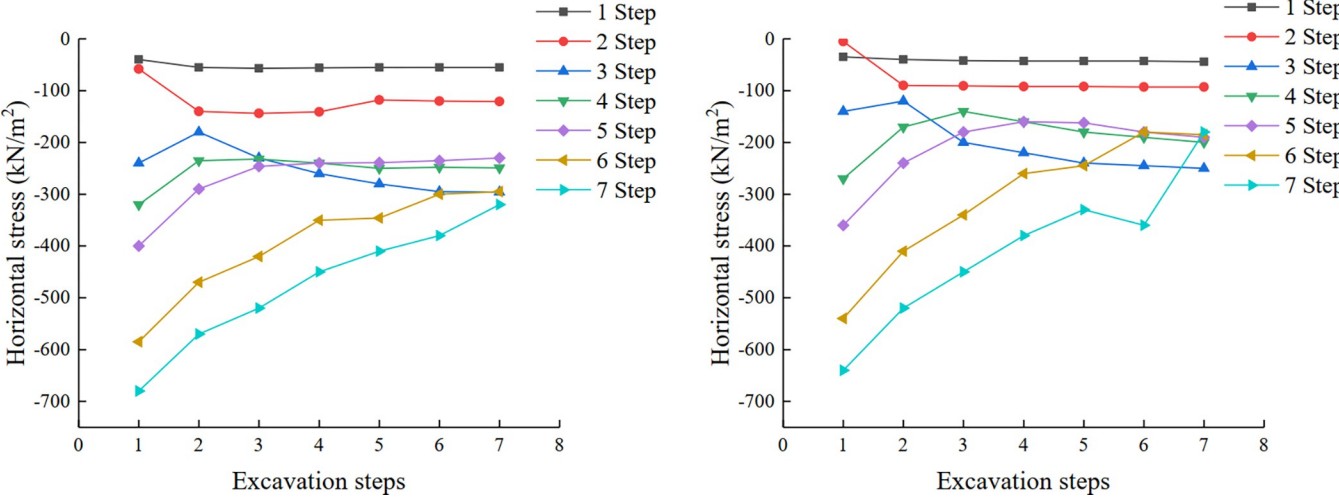

**Fig 9.** (a). The curve of horizontal stress variation of each step during slope excavation (in the middle of slope). (b). The curve of horizontal stress variation of each step during slope excavation (at the top of the slope).

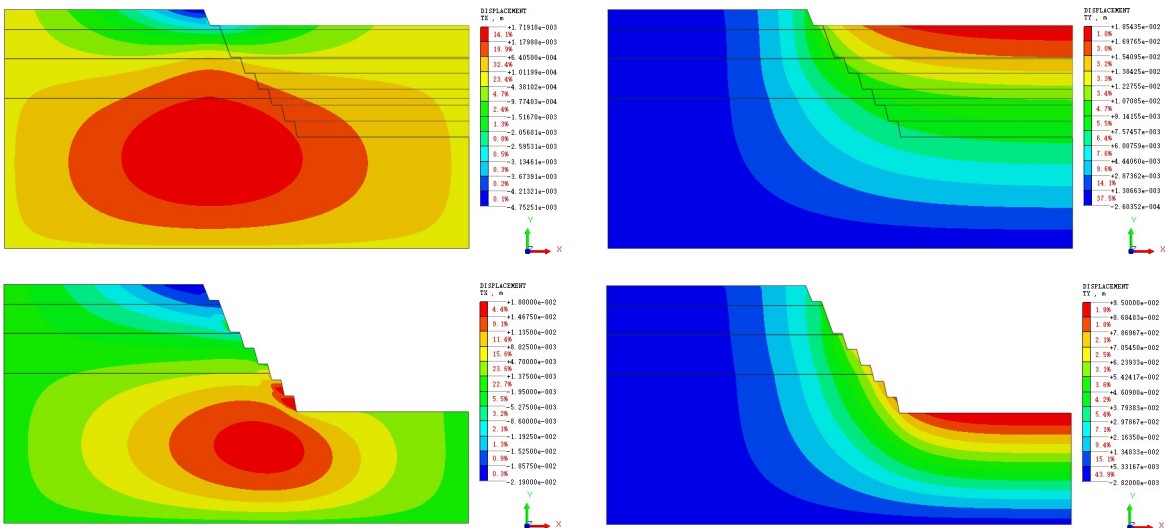

**Fig 10.** (a). Horizontal displacement cloud map of step 1. (b). Horizontal displacement cloud map after excavation. (c). Vertical displacement cloud map of step 1. (d). Vertical displacement cloud map after excavation.

According to the simulation results of schemes 3~4, the distribution rules of horizontal displacement, vertical displacement, and vertical stress of the slope are consistent with the schemes mentioned above. However, the distribution of horizontal tensile stress in schemes 3~4 gradually shifts to the upper part of the slope. With the change in slope angle in scheme 3 and 4, the plastic zone at the foot of the slope of step 3 and 4 continues to decrease, and the plastic zone at the foot of the slope of step 5 to 7 continues to expand to the top of slope, while the plastic zone at the foot of the slope of step 1 and 2 are gradually connected.

As shown in Table 6 and Fig 16, the slope stability coefficient gradually decreases with the increase in slope excavation depth and slope angle. When the first five steps are excavated, the stability coefficients of schemes 1 to 2 are all greater than 1.3, and the reduction of stability coefficients of the last two steps is also limited after the excavation is completed. Considering the control of the maximum extent of mining and the calculated results of the strength reduction method in Midas GTS finite element software, the slope angle of scheme 2 can be considered as the limit range temporarily. With the increase in the overall slope angle in scheme 3

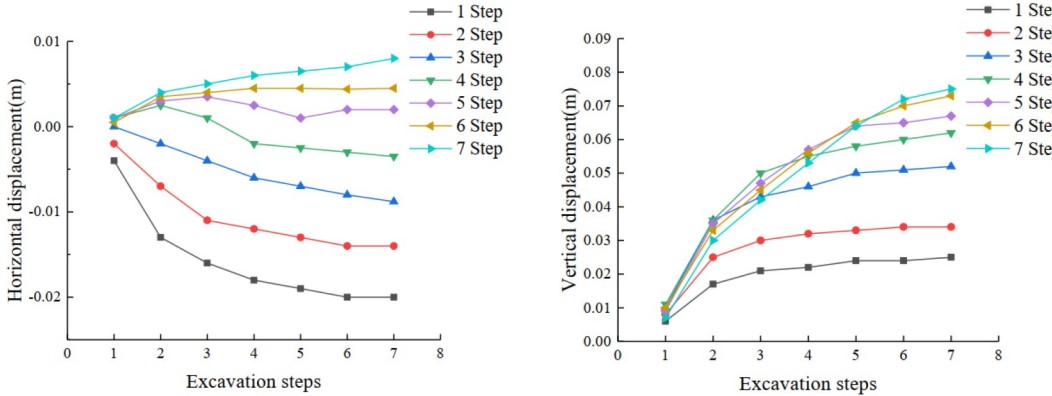

**Fig 11.** (a). Displacement curves of each step during excavation (horizontal displacement). (b). Displacement curves of each step during excavation (vertical displacement).

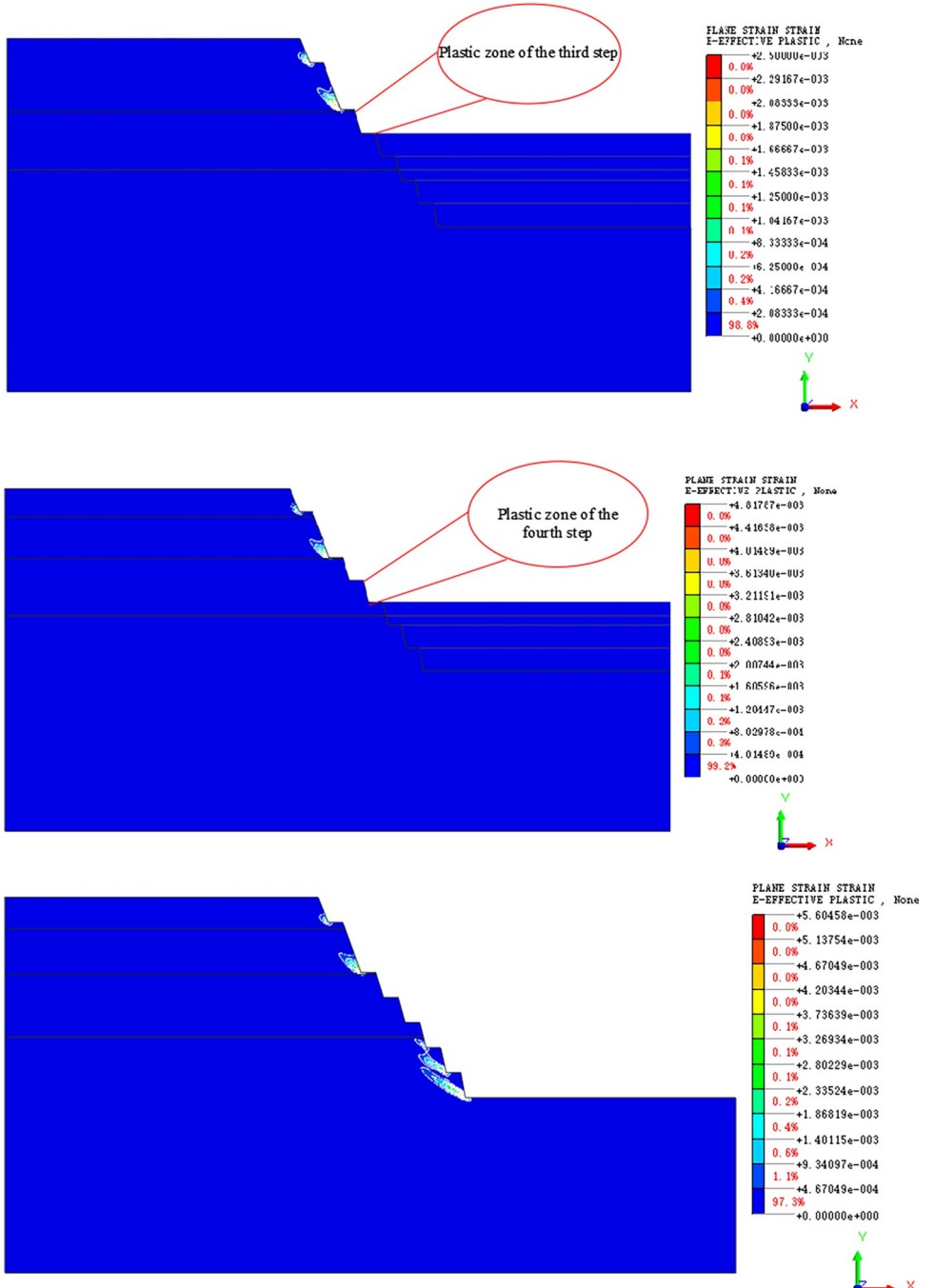

**Fig 12.** (a). Plastic zone at step 3. (b). Plastic zone at step 4. (c). Plastic zone after excavation.

and 4, the stability coefficient also shows a trend of gradual decrease. However, the final safety coefficient of scheme 3 and 4 decreases to around 1.3, and the stability coefficient of scheme 4 is lower than 1.3, so it is not considered. Considering that the stability coefficient obtained by the Midas GTS finite element software tends to be safe and combined with the above mentioned, the combination range of the optimal slope step slope angle is finally determined to be intermediate to scheme 2 and scheme 3.

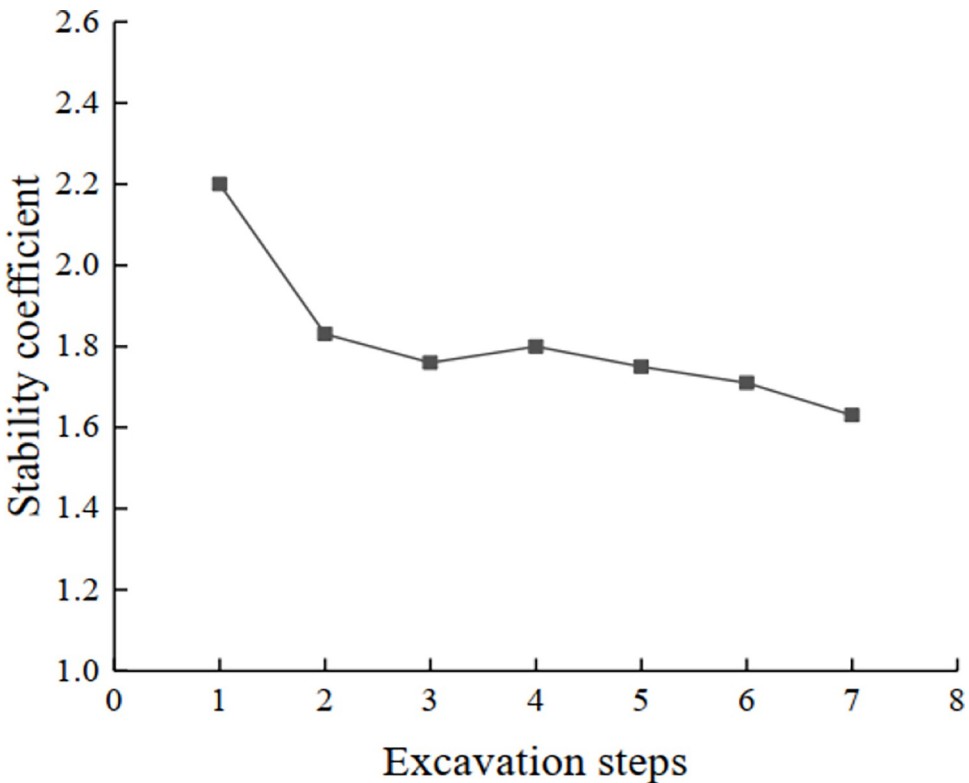

**Fig 13. Variation curve of slope stability coefficient.**

## Discussion

### Stability analysis of loading during excavation in the deep concave mine slope

The Midas-GTS software was used to analyze the stability variation. The results show that the horizontal stress increases gradually during the excavation process, and the stress concentration occurs at the foot of each step slope. In the process of excavation, the horizontal displacement increases gradually, and an obvious unloading rebound phenomenon occurs at the

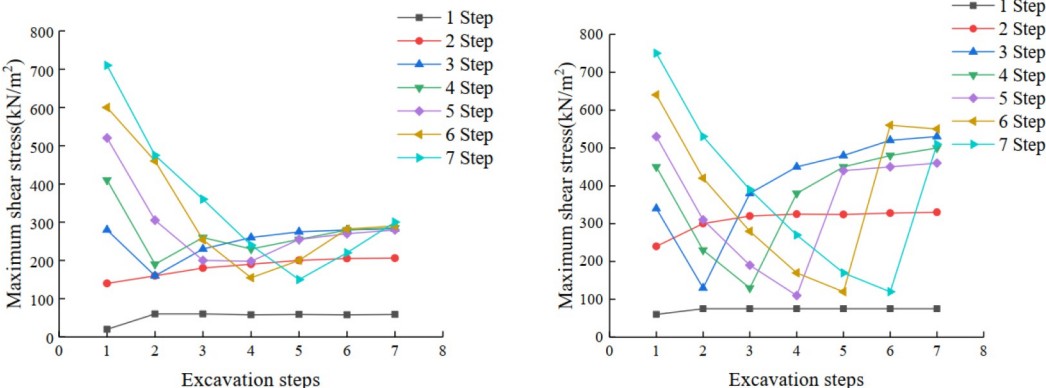

**Fig 14.** (a). Maximum shear stress curve of each step during slope excavation (in the middle of slope). (b). Maximum shear stress curve of each step during slope excavation (at the top of the slope).

**Table 4. Calculation of horizontal displacement of 1780-1768m stair under different excavation conditions.**

| Excavation steps (m) | Horizontal displacement (m) | | | | Variation value of horizontal displacement (m) | | |
|---|---|---|---|---|---|---|---|
| | Scheme 1 | Scheme 2 | Scheme 3 | Scheme 4 | Scheme 1–2 | Scheme 2–3 | Scheme 3–4 |
| 1826 -1816 | 0.0007 | 0.0007 | 0.0007 | 0.0007 | 0 | 0 | 0 |
| 1816 -1792 | 0.0017 | 0.0015 | 0.0016 | 0.0016 | -0.0002 | 0.0001 | 0 |
| 1792 -1780 | 0.0008 | 0.0004 | 0.0006 | 0.0006 | -0.0004 | 0.0002 | 0 |
| 1780 -1768 | -0.0001 | -0.0004 | -0.0001 | 0.0000 | -0.0003 | 0.0003 | 0.0001 |
| 1768 -1756 | -0.0011 | -0.0013 | -0.0008 | -0.0006 | -0.0002 | 0.0005 | 0.0001 |
| 1756 -1744 | -0.0013 | -0.0014 | -0.0007 | -0.0002 | -0.0001 | 0.0007 | 0.0005 |
| 1744 -1732 | -0.0008 | -0.0009 | 0.0001 | 0.0009 | -0.0001 | 0.001 | 0.0008 |

bottom of the pit. With the progress of excavation, the plastic zone of the slope develops continuously, and the stability coefficient decreases. This is consistent with the phenomenon of local landslide in the process of deep concave mine slope excavation.

## Analysis of optimum excavation slope angle of deep concave mine slope

The analysis of the experimental scheme shows that the stability coefficient decreases with the increase in excavation depth and slope angle. The stability coefficient of scheme 1 and 2 is relatively high, but considering the control of maximum mining amount and the calculation results of Midas-GTS software are biased towards safety, scheme 2 is relatively better. The stability coefficient of scheme 3 and 4 fluctuates around 1.3, and the stability coefficient of scheme 4 is significantly lower than 1.3. Therefore, scheme 3 is relatively better. Therefore, the optimum excavation slope angle is between scheme 2 and scheme 3.

**Table 5. Calculation of horizontal displacement of 1756-1744m stair under different excavation conditions.**

| Excavation steps (m) | Horizontal displacement (m) | | | | Variation value of horizontal displacement (m) | | |
|---|---|---|---|---|---|---|---|
| | Scheme 1 | Scheme 2 | Scheme 3 | Scheme 4 | Scheme 1–2 | Scheme 2–3 | Scheme 3–4 |
| 1826 -1816 | 0.0010 | 0.0011 | 0.0011 | 0.0011 | 0.0001 | 0 | 0 |
| 1816 -1792 | 0.0039 | 0.0040 | 0.0041 | 0.0041 | 0.0001 | 0.0001 | 0 |
| 1792 -1780 | 0.0051 | 0.0053 | 0.0054 | 0.0053 | 0.0002 | 0.0001 | -0.0001 |
| 1780 -1768 | 0.0057 | 0.0061 | 0.0061 | 0.0060 | 0.0004 | 0 | -0.0001 |
| 1768 -1756 | 0.0055 | 0.0060 | 0.0060 | 0.0055 | 0.0005 | 0 | -0.0005 |
| 1756 -1744 | 0.0091 | 0.0096 | 0.0109 | 0.0131 | 0.0005 | 0.0013 | 0.0022 |
| 1744 -1732 | 0.0114 | 0.0125 | 0.0151 | 0.0183 | 0.0011 | 0.0026 | 0.0032 |

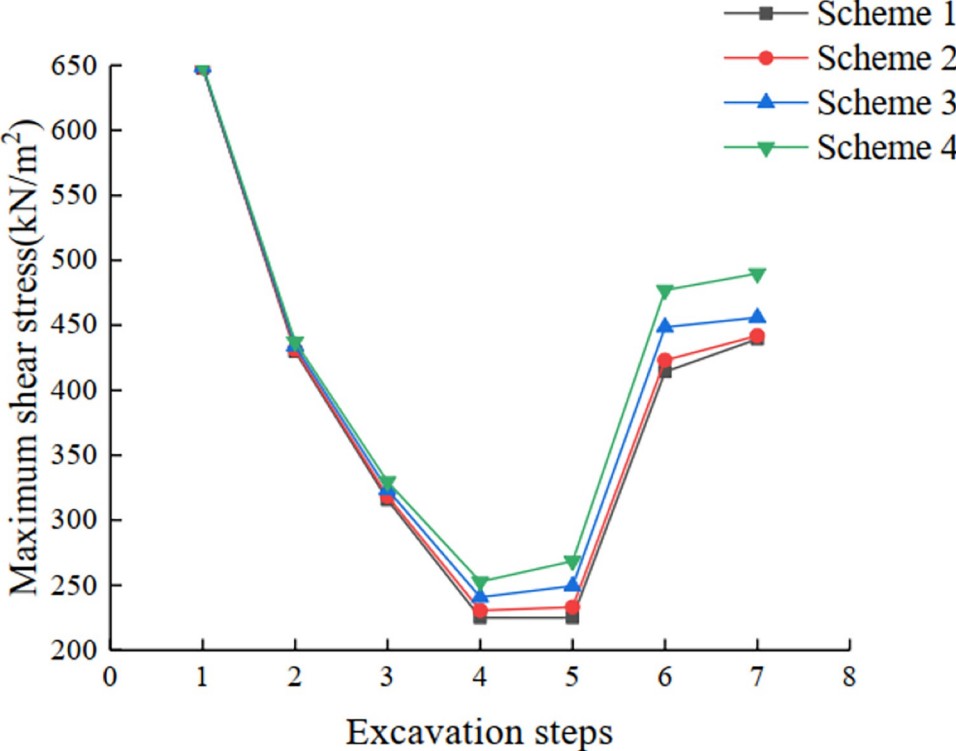

**Fig 15. Curves of maximum shear stress in different experimental schemes.**

## Research significance and limitations

This study focused on two aspects: stability analysis of excavation loading and optimal selection of excavation slope angle. Through theoretical research and numerical simulation analysis, it is found that the analysis results are consistent with the stability changes of the deep concave mine slope during the actual excavation process. The results of this study can provide a reference for future work related to the deep concave mine slope excavation. However, this study explored the stability changes solely in terms of anthropogenic engineering, without consideration of other geological and natural factors. Therefore, further research on the mining work with complex geological conditions is needed.

**Table 6. Stability coefficients of schemes 1 to 4.**

| Excavation steps (m) | Scheme 1 | Scheme 2 | Scheme 3 | Scheme 4 |
|---|---|---|---|---|
| **1826–1816** | 2.0816 | 2.1094 | 2.0500 | 2.0313 |
| **1816–1792** | 1.5781 | 1.5891 | 1.5156 | 1.5262 |
| **1792–1780** | 1.5813 | 1.5563 | 1.5039 | 1.5031 |
| **1780–1768** | 1.6004 | 1.5563 | 1.5531 | 1.4656 |
| **1768–1756** | 1.5688 | 1.5441 | 1.5156 | 1.4750 |
| **1756–1744** | 1.4820 | 1.4006 | 1.3953 | 1.3898 |
| **1744–1732** | 1.3719 | 1.3570 | 1.3313 | 1.2875 |

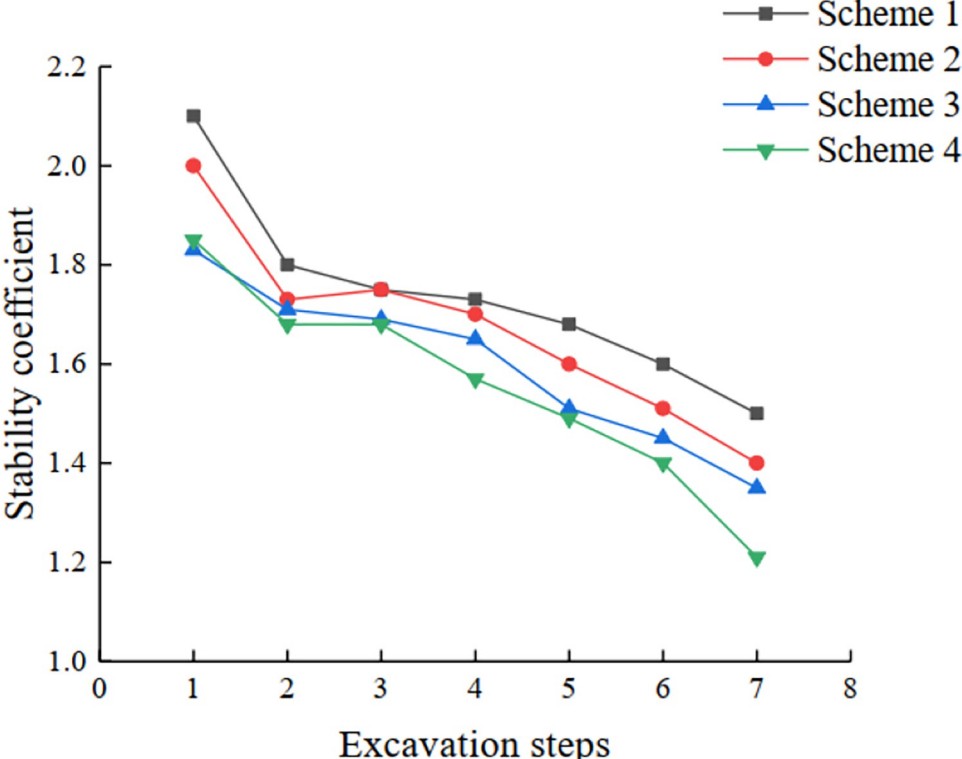

**Fig 16. Curves of stability coefficients for schemes 1 to 4.**

## Conclusions

This study aimed to investigate the disaster law in the process of mining slope excavation, and further explore the optimal selection of excavation angle. The main conclusions of the study are as follows.

1. The excavation loading process of the mine slope is a one-way process of reducing stability. The variation of excavation height ($H_1$-$H_2$) is the main dynamic factor that leads to the increase in the sliding force of the mine slope, which indicates that the process of excavation loading is a subjective factor that affects the stability of this kind of slope.

2. As the mining of the deep concave mine slope progresses, both the horizontal stress and vertical stress increase gradually. The horizontal displacement gradually changed from negative to positive. The plastic zone appeared at the foot of the step slope and gradually expanded upward.

3. With the increase in slope angle, the stability coefficient of the slope decreases to about 1.30 under the four experimental schemes. A small recovery in the safety factor in individual scenarios is caused by the stress redistribution and re-closure of existing fractures or shear bands. For example, the stability coefficients of the 1792–1780 step in scheme 1 rose from 1.5813 to 1.6004; the stability coefficients of 1792–1780 step in scheme 3 rose from 1.5039 to 1.5531.

4. According to the results of the design scheme and experiment scheme, the optimal value range of the slope angle of each step is determined between scheme 2 and scheme 3 as 62°~ 65°,64°~ 67°,67°~ 69°,70°~ 71°,73°,75°~ 76°,and 77°~ 80°.

## Supporting information

**S1 Fig. Schematic diagram of the excavation effect.** The essence of excavation effect is to remove the load in one or two directions from the slope originally in the three-way stress state and change its original stress state.
(TIF)

**S2 Fig. Stress distribution during excavation of rock-soil slope.** The solid line refers to principal stress trace, red dotted line refers to shear stress trace. The stress variation characteristics in the figure are as follows: (1) As the excavation goes on, the stress changes increase with the decrease of the distance from the excavation surface. (2) The shear stress zone is formed at the foot of slope, which is prone to failure.
(TIF)

**S1 File. Information about the model.**
(DOCX)

## Acknowledgments

The authors are grateful to anonymous reviewers for their valuable suggestions.

## Author Contributions

**Conceptualization:** Lili Wu, Keqiang He, Lu Guo, Linna Sun.

**Data curation:** Lili Wu, Lu Guo, Linna Sun.

**Funding acquisition:** Lili Wu.

**Methodology:** Lili Wu, Keqiang He.

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
