## [Decision Letter · Decision Letter 0]

10 May 2022

PONE-D-22-08272Analysis of stability law and optimization of slope Angle during excavation of deep concave mine slopePLOS ONE

Dear Dr. Wu,

Thank you for submitting your manuscript to PLOS ONE. After careful consideration, we feel that it has merit but does not fully meet PLOS ONE’s publication criteria as it currently stands. Therefore, we invite you to submit a revised version of the manuscript that addresses the points raised during the review process.

Dear Dr. Wu,

please find attached reviewers' remarks. Please read those carefully and address all the remarks.

Marko Čanađija

We look forward to receiving your revised manuscript.

Kind regards,

Marko Čanađija

Academic Editor

PLOS ONE

Journal Requirements:

Reviewers' comments:

Reviewer's Responses to Questions

**Comments to the Author**

1. Is the manuscript technically sound, and do the data support the conclusions?

Reviewer #1: Partly

Reviewer #2: Yes

2. Has the statistical analysis been performed appropriately and rigorously? 

Reviewer #1: No

Reviewer #2: Yes

3. Have the authors made all data underlying the findings in their manuscript fully available?

Reviewer #1: Yes

Reviewer #2: Yes

4. Is the manuscript presented in an intelligible fashion and written in standard English?

Reviewer #1: No

Reviewer #2: No

5. Review Comments to the Author

Reviewer #1: The paper is good but need effort to redraf since need more pay attention during writing. Analysi concept nee to brief clearly. Abstract nee to improved with consist : Background, objective, material & method, analysis result, conculsion.

novelty and limitiation of the research shall be brief at the manuscripts.

Other comment can be seen attachmnt. Goodluck

Reviewer #2: The sliding force and slope deformation behaviours during slope excavation are investigated in this work. The impact of various excavation slopes angle on slope stability under particular step slope height and width conditions is focused on. Generally, this manuscript needs to be substantially improved before it accepts. The detailed comments are following:

1) The manuscript needs extensive revision for language and grammar in two aspects. a) a translation agency or computer program is required to improve general English; b) The manuscript must be reviewed by a native English speaker so that readers can have a clear understanding of the goals and results of the research. Some strange phrases occur throughout the manuscript. All related unprofessional words cannot be pointed out. Here are only some examples.

In Abstract, “law of disaster”, “law of stability”, “provide reference”

“Influence law” in Line 74, “sliding power” in Line 81,

2) Why “slope Angle” is in an uppercase form throughout the text?

3) The Abstract should be shortened and modified: the abstract should contain Objectives, Methods/Analysis, Findings, and Novelty /Improvement.

From your Abstract “this paper analyzes ….; This paper takes….; this paper analyzes”.

Please summarize your Methods/Analysis.

4) In Introduction, since the numerical analysis for the excavation of deep concave mine slope is one of the important parts of this manuscript, previous research on the numerical methods should be introduced in Introduction in a single chapter. However, this related information can be found in Introduction.

5) Line 79, stress redistribution is the typical phenomenon and process during the excavation-induced slope failures. Some related references on stress redistribution should be added. Fang K, Miao M, Tang H, Dong A, Jia S, An P, Zhang B and Tu J (2022) Model test on deformation and failure behaviour of arching-type slope under excavation condition. Eng Geol: 106628. Fang K, Tang H, Su X, Shang W and Jia S (2020) Geometry and maximum width of a stable slope considering the arching effect. Journal of Earth Science 31: 1087-1096.

6) Lines 88-95, the notation should be changed in Fig. 1 instead of the text. Is “KN” the unit of each parameter? If so, “KN” should be “kN”. “KPa” should be “kPa” in line 108 and other places.

7) Line 175, in this study, the Midas-GTS software is applied. The advantages of this software should be clarified. In other words, suitable comparisons between other software should be added.

8) The discussions about the optimum slope angle during the excavation with other related research should be added in the new section Discussion before Conclusion.

9) In Reference, some references should be marked with the original language. For example,

32. Ma XY. Study on the excavation unloading dynamic effect and stability evolution rule of the open-pits rock slope. Qingdao Technological University. 2014. (in Chinese)

10) The figures in this paper must be substantively improved. For example,

What’s the meaning of the blue part in Fig. 2?

Where is the location of the fault and related to any slopes?

Add the corresponding elevation in Fig. 5.

Scales should be added in Fig. 6. The crack is not clear in Fig. 6(a). Please draw some lines to make it clear.

The minor types of tick marks in Fig. 14 and in similar figures are not necessary.

6. PLOS authors have the option to publish the peer review history of their article (what does this mean?). If published, this will include your full peer review and any attached files.

Reviewer #1: No

Reviewer #2: No

---

## [Author Response · Author response to Decision Letter 0]

16 Jun 2022

Dear reviewers and editor:

I am very grateful for your efforts and comments for the revisions of our manuscript. According to your comments and suggestions, the revisions on the manuscript ' Analysis of stability law and optimization of slope angle during excavation of deep concave mine slope' (No. PONE-D-22-08272_R1) have just been completed.

The responds to your comments and the main corrections in the manuscript are as flowing:

(Reviewer #1:)

1. The abstract of the paper has been redrafted, consisting the background, objective, material& method, and results with all not more than 250 words.

2. Detailed information on the characteristics of the materials have been provide in Table 1 and Table 2.

3. The tables in the paper have been checked and improved.

4. The consistency type of word has been used in paper.

5. The conclusions have been simplified and some numbers have been added to conclusions for quantitative analysis.

6. The relevant numbers have been described in conclusion 2.

7. Conclusion 3 has been quantitatively stated based on the research results.

8. The numbers in Fig 8(b) are correct. This figure is the vertical stress cloud diagram after the mine slope excavation is completed. As a result of excavation unloading, the vertical compressive stress at the foot of slope gradually decreases and tends to variety to the horizontal tensile stress. Macroscopically, the phenomenon is the rebound of excavation unloading.

9. In the process of numerical simulation, the author has considered how to reflect the time effect, but considering that the more mature time history application engineering in MIDAS-GTS software is earthquake engineering, the time history analysis in slope engineering needs to be further explored. In the future study, the author will continue to consider using time to change excavation steps, hoping to obtain some results.

10. The plasticity of material is indeed related with increasing plasticity on the model. However, the plastic zone range is very small at the third and fourth steps of excavation in Fig 13, which is due to the closure of cracks in the slope or the reduction of shear failure caused by dynamic excavation. It can also be seen from the stability coefficient that the stability of the third and fourth steps has rebounded slightly, which is more stable than the first two steps.

11. The answer to this question is similar to the answer to question 9.

12. In Fig14 (b), when the excavation of each step is completed, the shear stress at the slope toe of each step shows an increasing trend. This is because the overall height of the slope increases after the excavation of each step, and the possibility of shear failure of the slope increases, which is consistent with the law of deformation and failure of the slope.

(Reviewer #2:)

1. The manuscript has undergone extensive revision for language and grammar in two aspects.

2. The “slope Angle” has been changed to lowercase form throughout the text.

3. The abstract has been modified, and contained the objectives, methods, findings and novelty. 

4. Previous researches on the numerical methods have been introduced in Introduction in a single chapter.

5. Some related references on stress redistribution have been added.

6. The text in Fig 1 has been deleted; The parameter unit has been changed from KN to kN and KPa to kPa.

7. The advantages of Midas-GTS software and suitable comparisons between other software have been added to the article.

8. The discussions about the optimum slope angle during the excavation with other related research have been added in the new section Discussion before Conclusion.

9. In Reference, some references have been marked with the original language.

10. The figures in this paper have been substantively improved.

 (1) The meaning of the blue part in Fig 2 is Gansu Province.

 (2) The fault is located near the platform 1732 and is part of the Jinchuan deep concave mine slope; Fig 3 has been modified.

 (3) The corresponding elevation has been added in Fig 3.

 (4) Scales and some lines have been added in Fig 5.

 (5) The minor types of tick marks in Fig 13 and in similar figures have been deleted.

The novelty and limitation of the research have been briefed at the manuscripts.

All revisions in detail are shown in the revised manuscript with changes marked in blue. We appreciate for you warm work earnestly, and hope that the correction will meet with approval.

Once again, thank you very much for your comments and suggestions.

Sincerely yours

Lili Wu

---

## [Decision Letter · Decision Letter 1]

6 Jul 2022

Analysis of stability law and optimization of slope angle during excavation of deep concave mine slope

PONE-D-22-08272R1

Dear Dr. Wu,

We’re pleased to inform you that your manuscript has been judged scientifically suitable for publication and will be formally accepted for publication once it meets all outstanding technical requirements.

Kind regards,

Marko Čanađija

Academic Editor

PLOS ONE

Additional Editor Comments (optional):

Reviewers' comments:

Reviewer's Responses to Questions

**Comments to the Author**

1. If the authors have adequately addressed your comments raised in a previous round of review and you feel that this manuscript is now acceptable for publication, you may indicate that here to bypass the “Comments to the Author” section, enter your conflict of interest statement in the “Confidential to Editor” section, and submit your "Accept" recommendation.

Reviewer #1: All comments have been addressed

Reviewer #2: All comments have been addressed

2. Is the manuscript technically sound, and do the data support the conclusions?

Reviewer #1: Yes

Reviewer #2: Yes

3. Has the statistical analysis been performed appropriately and rigorously? 

Reviewer #1: No

Reviewer #2: Yes

4. Have the authors made all data underlying the findings in their manuscript fully available?

Reviewer #1: Yes

Reviewer #2: Yes

5. Is the manuscript presented in an intelligible fashion and written in standard English?

Reviewer #1: Yes

Reviewer #2: Yes

6. Review Comments to the Author

Reviewer #1: (No Response)

Reviewer #2: The authors have addressed all the issues raised by the referee. This version can be accepted for publication.

7. PLOS authors have the option to publish the peer review history of their article (what does this mean?). If published, this will include your full peer review and any attached files.

Reviewer #1: No

Reviewer #2: No

---

## [Editor Report · Acceptance letter]

13 Jul 2022

PONE-D-22-08272R1 

Analysis of stability law and optimization of slope angle during excavation of deep concave mine slope 

Dear Dr. wu:

I'm pleased to inform you that your manuscript has been deemed suitable for publication in PLOS ONE. Congratulations! Your manuscript is now with our production department. 

Kind regards, 

on behalf of

Dr. Marko Čanađija 

Academic Editor

PLOS ONE